# Implementation of the WHO guideline on treatment of young infants with signs of possible serious bacterial infection when hospital referral is not feasible in rural Zaria, Nigeria: Challenges and solutions

**Robinson Daniel Wammanda**[1]*, **Shadrach Aminu Adamu**[2], **Hyellashelni Daba Joshua**[3], **Yasir Bin Nisar**[4], **Shamim Ahmad Qazi**[5], **Samira Aboubaker**[5], **Rajiv Bahl**[4]

1 Department of Paediatrics, Ahmadu Bello University and Ahmadu Bello University Teaching Hospital, Zaria, Nigeria, 2 General Hospital Kawo and Kaduna State Ministry of Health, Kaduna, Nigeria, 3 Ahmadu Bello University Teaching Hospital Tudun Wada, Zaria, Nigeria, 4 Department of Maternal, Newborn, Child and Adolescent Health, World Health Organization (WHO), Geneva, Switzerland, 5 Retired from WHO, working as a WHO consultant, Geneva, Switzerland

* rwammanda@gmail.com

## Abstract

### Background

Bacterial infection is one of the leading causes of mortality in young infants globally. The standard practice to manage young infants with any sign of possible serious bacterial infection (PSBI) is in a hospital setting with parenteral antibiotics, which may not be feasible for majority of cases in most low resource settings. The World Health Organization developed a guideline on management of PSBI in young infants when referral is not feasible in 2016.

### Methods

We conducted implementation research in selected communities in Zaria Local Government Areas of Kaduna State with an estimated population of 50,000 with the aim of understanding how to implement the WHO PSBI treatment guideline to achieve high coverage with low case fatality and treatment failure rates. Implementation was within the programmatic settings using existing health structure. We conducted policy dialogue with decision makers to adapt the recommendations to their social, cultural and programmatic context in Nigeria, held orientation meetings with program managers, built capacity of the health workers and supported the implementation within the health system. We supported a non-government organization to conduct community sensitization to promote care seeking and adherence to treatment advice. The research team collected data systematically on all young infants identified to have PSBI, the treatment they received and the clinical outcome.

### Results

Between April 2016 and March 2017, we identified 347 young infants up to 2 months of age with signs of PSBI who received treatment either as an outpatient or in a hospital among

**Data Availability Statement:** All relevant data are within the manuscript and its supporting information files.

**Funding:** The study was funded by the Bill and Melinda Gates Foundation through a grant to the World Health Organization. The funder had no role in study design, data collection and analysis, decision to publish, or preparation of the manuscript.

**Competing interests:** The authors have declared that no competing interest exist.

2,154 births in the study population. The coverage of PSBI treatment in the study area was 95.5% assuming that 10% of all births have an episode of PSBI in the first two months of life. Most (89%) sick young infants with PSBI were identified by the community-oriented resource persons and sent to the Primary Health Care Centres (PHCs). Most families (97%) refused referral and were treated at a primary health care centre on outpatient basis. There were 12 deaths (3.5%) and 17 non-death treatment failures (4.9%) in 343 infants in whom an outcome could be ascertained. While non-death treatment failure rate was highest in 0-6-day infants with fast breathing (14.4%), case fatality was highest in those with signs of critical illness (20%).

## Conclusion

We have demonstrated that outpatient treatment strategy for young infants with PSBI when referral is not feasible is implementable within the programmatic settings, achieving very high population coverage and relatively low treatment failure and case fatality rates. Implementation at scale will require government's commitment to strengthen the health system with trained, motivated health care providers and necessary commodities.

## Introduction

Severe infection in the neonates commonly referred to as possible serious bacterial infection (PSBI) is a leading cause of mortality in young infants contributing to as much as 37% of the 2.6 million neonatal deaths globally[1–3] In sub-Saharan Africa, the incidence of PSBI has been estimated to be 7.6% with a case fatality rate of 9.8%.[4]In Nigeria, the AFRINEST study showed that over 10% of live birth will develop PSBI.[5–6]

The standard practice to manage young infants with any sign of PSBI is in a hospital setting with parenteral antibiotics, which may not be feasible in low resource settings, especially in some low- and middle-income countries (LMIC). Simplified regimens comprising of injectable plus oral antibiotics delivered outside the hospital setting when referral was not feasible, were shown to be effective by Bang et al in India,[7] Baqui et al in Bangladesh [8]and Zaidi et al in Pakistan [9]. Later, three large community-based trials were designed in settings where referral was not feasible in five African and Asian countries (Bangladesh, Democratic Republic of Congo, Kenya, Nigeria and Pakistan)[5–6, 10–11] Their results showed that barring critically ill young infants, simplified antibiotic regimens could safely and effectively treat young infants with signs of PSBI when referral was not feasible.

The above evidence contributed to the development of the World Health Organization (WHO) guideline for the management of young infants with signs of PSBI when referral is not feasible. [12]In order for this guideline to be implemented, individual countries needed to review the guideline, adapt it to suit their local socio-economic, cultural and health systems contexts and make policy decisions for implementation within their programme settings. In Nigeria, a policy dialogue with the Ministry of Health and other stakeholders highlighted the need for implementation research to learn how to implement the guideline in a programme setting.

We report the results from implementation research that was designed to inform on the use of simplified management of sick young infants with signs of PSBI at first level health care facilities when referral is not feasible within the programme setting in Zaria, Nigeria.

## Methodology

### Study setting

**Characteristics of the study site.** The implementation research was conducted in Zaria Local Government Area (LGA), located in the northern part of Kaduna State in the North West geo-political zone of Nigeria. A policy dialogue at the central level was held in Abuja, in which State and local government functionaries and other technical experts took part. It was decided that the national policy should be adapted to implement WHO PSBI guideline when referral is not feasible in two places in Nigeria, one in the North part of the country and one in the South to represent both diverse regions. In the North part, Kaduna state was proposed and in South part, Oyo State was proposed. Zaria LGA is representative of a population where referral is not accepted by many families with sick young infants. We also held consultations with the local authorities and in consultation chose certain wards in Zaria LGA to implement this guideline.

Zaria LGA is the headquarters of Zazzau Emirate and the location of the traditional ancient city of Zaria. It is made up of more than 100 urban, peri-urban and rural settlements populated by predominantly Hausa/Fulani Moslems. The LGA has a population of 300,000.[13] The LGA is divided into 11 political wards. Each settlement has a village head who is appointed by the Emir of Zaria, the paramount ruler of Zazzau Emirate.[14] The status of the women in the LGA and other parts of northern Nigeria is comparatively low, with limited access to formal education, marginalization in decision making, even in matters of health decision-making, economic dependence and limitation of movement and work through the widespread practice of female seclusion being the norm. Husbands and traditional/religious leaders are important gatekeepers, while mother in-laws and co-wives are very significant others; all these people play important role in health seeking decisions.[15] The main profession of the people in these communities is subsistence farming.

The infant mortality and neonatal mortality rates in Nigeria are 67/1000 and 39/1000 live-births, respectively.[16] Local studies in communities in Zaria LGA have shown that the LGAs have a higher than national crude birth rates ranging between 45-50/1000 population. [17] As it is for other parts of Kaduna state, the coverage for health intervention such as basic vaccination is poor, with only 20% of children 12–23 months receiving all basic vaccination at any time during the 2018 National demographic and health survey.[16]

In these communities, more than half of the pregnant women attend antenatal clinic but most deliveries take place at home with family members in attendance.[15,18] While the Traditional Birth Attendants play a limited role during pregnancy, they play a major role during delivery and baby care until the time of naming ceremony usually by the seventh day after birth. Home treatment, traditional medicine and use of patent medicine vendors are usual first line actions in the event of ill health for the child.

The research was carried out in two rural political wards of Zaria LGA covering over 50 communities scattered over variable distances from their nearest Primary Health Care centres (PHCs). These two political wards were chosen as they are among the most rural of these political wards and with limited accessibility to referral centers. Some communities are up to 15 kilometres away from the nearest health facility. The selected catchment areas for the implementation research had an estimated population of under 50,000 and 2,500 births per annum. We expected about 250 PSBI cases per year as seen in the AFRINEST study.[5]

**Health system at the study site.** Primary health care is provided through PHCs, which serve a population of 5000–6000, where patients self-present. The PHCs are primarily staffed by nurses, Community Health Extension Workers (CHEWs) and midwives. Some PHCs are only staffed by CHEWs. General hospitals provide secondary health care services and are

staffed with medical doctors and nurses. In some general hospitals, there may be specialist doctors trained in pediatrics or neonatology and a newborn nursery.

Medicines are in principle supplied to the PHCs by LGA through the Health Department of the LGA. At the time of this research, Kaduna State Government operated a free Maternal and Child Health service as a policy. However, the state also operated a Sustainable Drug Supply System (SDSS), whereby the State Government purchased drugs as seed drugs and distributed to the various LGAs. "Seed drug" is a one-off free bulk supply of essential medicine by the state government to the LGAs or PHCs that is expected to be replenished by the funds generated from the sale of the drugs. The LGAs in turn supplied the PHCs with a seed drug supply. The PHC staff dispensed these drugs to patients on cash, which was used to replenish their stock from the LGA stores. The LGA in turn used the money generated to replenish their stock from the State central medicines store.

**Status of management of sick young infants.**   Although Integrated Community Case Management (iCCM)[19–20]as a national strategy was operational in Nigeria, this had not been introduced in communities within Zaria LGA. Families often sought care for their sick young infants from traditional healers, chemists or the nearby PHC. At the PHCs, health workers in general referred sick young infants to either Hajiya Gambo Sawaba General Hospital, Zaria or the Ahmadu Bello University Teaching Hospital, Shika at a distance of about 25 kilometres. Transportation is limited and access to referral level facilities is poor.

## Phases of implementation

We adapted the RE-AIM framework [21] for planning and evaluation of this study. Fig 1 explains the conceptual framework used in this study based on the RE-AIM framework.

**National level policy dialogue for adoption of WHO PSBI guideline.**   Federal Ministry of Health (MOH) with support from WHO organized a national level orientation and policy dialogue with national and state level stakeholders in Abuja in March 2015 to review the WHO guideline and make informed decision about its implementation in Nigeria. Policy makers, programme implementers within the MOH and non-governmental organizations (NGOs) as well as the paediatric community participated in the meeting. The meeting created a platform for dialogue and for decisions on small scale implementation for gaining experiences and insights into policy requirements. Key issues were raised and collective decisions were taken. Details of these decisions were conveyed to WHO through an official letter (FHD/ CH/3054/1) dated 23rd April 2015 (Table 1). Fig 2 provides definitions of health workers category, setting in which they practice and the scope of their practices.

**Local level dialogue with the health authorities and the community.**   We held a meeting with the Zaria LGA authorities and officials of the health department at the local government secretariat on the 18th September 2015. We shared the outcome of the national orientation and policy dialogue and the decisions about PSBI management and the proposal to conduct the implementation research in the PHCs of two wards, Wucicciri and Dutsen Abba, these being the most rural of the political wards in the LGA. The authorities welcomed the initiative and agreed with the proposal and with the strategy which included involvement of health workers already working at the PHCs in the research.

We organized another orientation meeting with the community leaders for the two political wards. The meeting was conducted in the compound of the ward district leaders within the communities respectively on the 6th and 7th of October 2015. The participants in this meeting included district head of the wards, village Sarkis (village heads), the mai unguwas (sub-village heads), heads of households from various communities and the PHC staff. We described the purpose of the implementation research and potential benefits to the communities. As there

                                                           

**Fig 1. Conceptual framework for implementation research.** The implementation framework was based on the RE-AIM framework. RE-AIM, Research Effectiveness Adoption Implementation Maintenance.

were no community health workers in the community to help identify pregnancies and births, we discussed the selection of Community Oriented Resource Persons (CORP) from their communities, who would be young men or women who met a minimum education standard stipulated by the Federal Government policy. The CORPs would be hired to work in the community to identify pregnant women and births.

                                                                                          

**Table 1. Issues and decisions made during the national policy dialogue for management of PSBI in young infants 0–59 months of age when referral is not feasible.**

| Issues | Decision |
|---|---|
| Where will the treatment be provided? | At the #PHCs |
| Who will identify sick young infants in the community? | - State based CORPS/VHW/≠CHEWS)<br>- Not PPMVs |
| Where will the sick young infant be referred from the community? | To the PHCs |
| Who will confirm presence of fast breathing; or signs of PSBI? | CHEWs/Nurses/Midwives/CHOs who is available at PHCs will confirm the signs |
| Who will refer the young infants with signs of PSBI to the hospital? | CHEWs/Nurses/Midwives/CHOs who is available at the PHC will refer to the hospital |
| Who will provide treatment if referral to the hospital is not accepted by family for young infants with signs of PSBI? | - CHEWs/Nurses/Midwives/CHOs whoso ever is available will treat sick young infants with signs of fast breathing or clinical severe infection.<br>- Young infants with signs of critical illness will continued to be referred to a hospital |
| Which antibiotic regimen will be used for clinical severe infection? | Intramuscular injection gentamicin for 2 days and oral amoxicillin for 7 days. |
| Which antibiotic regimen will be used for young infants with fast breathing? | -Oral amoxicillin for 7 days for 7–59 days old young infants without referral to hospital<br>- If referral not accepted by family for 0–6 days old young infants, they will be treated with oral amoxicillin for 7 days |
| Set up implementation research sites | Ibadan (Oyo State) and Zaria (Kaduna State) |

Abbreviations

PHC; primary health care centre

CORPs; community-oriented resource persons

VHW; village health workers

Chews; community health extension workers

PPMV; patent private medicine vendors; (private individuals operating a chemist shop who are not qualified pharmacists)

CHO; community health officers

**Pre-implementation evaluation of existing PHCs, health workers and commodities.**
The implementation research was implemented within the existing health system of the state, so we assessed the status of health facilities and human resources in the selected wards to inform the implementation of the guideline using the modified version of "Equipment and supply checklist" from the WHO health facility survey.[22] In Dutsen Abba ward, there were three PHCs and in Wuciciri ward there were two PHCs. In one PHC there was no staff, one PHC had two Community Health Extension Workers (CHEWs) and others had only one CHEW. Some of the communities expected to be served by the PHC located at Wuciciri were too far away from the PHC to the extent of limiting accessibility by families of sick infants. The CHEWs posted at PHCs had received training in 2014 but did not have any refresher training recently. Essential medicines for the treatment of young infants with signs of PSBI (gentamicin, amoxicillin and ampicillin) when referral for hospital is not possible were not available routinely in the health facilities.

**Establishment of technical support unit and the demonstration site.** A technical Support Unit (TSU) was established to help set up the study site, collaborate and coordinate with the health authorities, providing training and technical support to the CORPs and PHC staff, monitor quality of implementation, ensure supplies, identify challenges and solutions to

**Nurse is a** health worker with formal training in schools of Nursing. They undergo 5 years of training leading to the award of Diploma is General Nursing. They are expected to spend 100% of their time at the health facilities providing general nursing care including administration injections and oral medicines.

**CHO** is a Community Health Officer. They are community health workers with additional training designed to train them to become the most senior of the Community Health Practitioners in Nigeria. They undergo two years programme leading to the award of Higher Diploma in Community Health. Thirty percent of the practitioner's time is spent in the Community while seventy percent is spent in the clinic.

**CHEW** is a Community Health Extension Worker. They are health workers with 3 years of formal training in schools of health technology leading to the award of Diploma in community health. The Community Health Extension Worker (CHEW) is a member of the health team for Primary Health Care (PHC). The Community Health Extension Worker will spend 50% of his time on Community based functions and 50% in the Clinic. Their clinic-based function among other include provision of integrated Primary Health Care Services, treat common conditions and injuries including administration of injectable and oral drugs.

**CORP** is a Community Oriented Resource Person. They are lay members of the community should be selected for training, by the community based on a set of criteria that has been developed with the input of representatives of the community. They provide health education on prevention and treatment; including serving as information resources within the communities, as well as dispense recommended anti-malarial, antibiotics, ORS/Zinc, commodities to caretakers of children under-five years of age with illnesses and to older members of the community in line with national guidelines.

**Fig 2. Definitions of health workers category, setting in which they practice and the scope of their practices.**

overcome them. The TSU members included the principal investigator, project manager, two field supervisors and six CHEWs (one each from the study PHCs). The project manager assisted the PI in the day to day coordination of activities, supervision of the study staff. The project manager was also trainer for Community Newborn Care and also contributed to the adaptation of the WHO Integrated Management of Childhood Illness (IMCI) training package for the training of CORPs and Nurses/CHEWs. The TSU addressed some of the pre-implementation issues.

**Implementation of the PSBI treatment guideline. Training of health workers:** Training on assessment, classification and management of young infants with PSBI signs when referral is not feasible using the young infant component of IMCI training package [12, 23] was conducted in two stages. Initially, Master Trainers from all the study sites were trained by WHO facilitators for three days between the 9[th] and 11[th] of November 2015.

Subsequently, the Master trainers provided onsite training to the Nurses and CHEWs in Zaria.

The training which was conducted onsite in Zaria, consisted of six days (15[th] to 19[th] February 2016) of interactive communication, clinical demonstrations and hands on skills practices in small groups. Trainees were taken to the Paediatric Department of the nearby hospitals to learn identification of signs of illness at patients' bedside. The training included sessions on assessment of treatment outcomes (i.e., clinical treatment failures including death, adherence to therapy to both gentamicin and oral amoxicillin and referral for hospitalization). A second training was an onsite 6- day training on home visits for care of the newborn and identifying sick young infants for CHEWs and CORPs conducted by the Master trainers supported by the study coordinators using the WHO training manual for care of the newborn at home.[24]

**Field supervision:** Two field supervisors supervised the PHC nurses/CHEWs and CORPs working in the community. They conducted quality assurance checks through standardization exercises, reviewed all completed study data collection forms by the nurses/CHEWs, cross checked drug dosages and administration by both nurse/CHEWs, and checked functional status of all working tools in the field. They also resolved problems encountered by the field staff on a daily basis. The Field Supervisors who had received a training of trainers' course during the AFRINEST study also served as trainers during the WHO/UNICEF Training course for Community Health Workers on the Caring for the Newborn at home and contributed to the adaptation of IMCI training for CORPs and nurse/CHEWs.

The Zaria LGA Monitoring and Evaluation officer worked with the field supervisors to effectively supervise the activities of the CORPs in the communities and Nurses/CHEWs in the PHCs.

**Identifying sick young infants in the community.** CORPs, identified by the communities were given an honorarium and transport allowance. They worked with the traditional birth attendants within their assigned communities to identify pregnancies and births. The CORPs registered pregnant women and conducted two home visits to promote antenatal care and delivery by a skilled birth attendant at the PHC as well as birth preparedness. The CORPs followed the newborns on day 1, 3, and 7 after birth to promote essential newborn care [24]and to assess signs of illness. Small babies were also visited on day 2. The newborns were followed for up to two months of age. Any young infant identified with a danger sign was referred to the nearby PHC for further assessment. If the families refused to take their sick young infant to the PHC, the CORPs invited the nurse/CHEWs to visit the young infant's home for further assessment. Traditional Birth Attendants were given transport and telephone incentives to facilitate movement and communication.

**Assessment and management of sick young infants at the PHC and referral to hospital when required.** The PHC nurse/CHEWs assessed sick young infants sent by the COPRs or brought by families for signs of PSBI using IMCI [17]. Those who required referral were referred to the Special Care Baby Unit (SCBU) of Ahmadu Bello University Teaching Hospital, Shika, Zaria. They were counseled about the need for hospitalization for their infant.

When families refused hospital referral despite the best efforts of the PHC nurse/CHEW, their sick young infants with PSBI signs were re-classified into either fast breathing pneumonia, clinical severe infection (CSI), severe pneumonia or critical illness (Fig 3).

Young infants 7–59 days of age with only fast breathing were treated with oral amoxicillin without referral. Infants with CSI were offered treatment with simplified antibiotic regimens on outpatient basis. Treatment was initiated after written informed parental consent was obtained. Young infants with critical illness were referred again, but if families still refused referral they were treated with intramuscular gentamicin and ampicillin daily while reinforcing referral.

Injections and first daily dose of oral medicine were administered by the PHC nurses/ CHEWs at the PHC or at home (if the infant was not brought for second injection). Second oral dose for the day was administered by mothers at their homes. For infants who vomited within 20 minutes of oral dosing, the caregivers were instructed to re-administer a complete dose. Under treatment, infants were evaluated daily by the same nurse/treating health worker who provided injectable treatment. CORPs visited the young infants who were on only oral therapy at home. All infants treated at home were followed up on days, 2, 4, 5, 7, and 14 after enrollment. On each follow-up visit, the infant was examined for signs of improvement or deterioration and for any adverse effects. Records of treatment received and follow up visits were documented by the treating nurse or CHEW. Parents were counseled on signs of deterioration and were asked to bring the infant back to the PHC if they noticed any sign. Infant whose condition deteriorated at any visit, was referred to the hospital for further management

**PSBI** is defined as young infants 0-59 days with any of the following signs: fast breathing (respiratory rate $\geq$ 60 breaths/minute), severe chest in-drawing, high body temperature $\geq$ 38 °C, low body temperature < 35.5 °C, no movement at all or movement only on stimulation, not able to feed at all or not feeding well and convulsions.

*Sub-Classifications and treatment*

- Pneumonia - Fast breathing (respiratory rate 60 or more per minute) as the only sign in infants 7-59 days old
  - Recommended treatment: oral amoxicillin twice daily for 7 days without referral

- Severe Pneumonia - Fast breathing (respiratory rate 60 or more per minute) as the only sign in infants 0-6 days old
  - Recommended treatment: Referral to the hospital. If referral not feasible treat with oral amoxicillin twice daily for 7 days
- Clinical Severe Infection – Not feeding well, movement only on stimulation, severe chest indrawing, high body temperature $\geq$38°C or low body temperature <35.5°C
  - Recommended treatment: Referral to the hospital. If referral not feasible injection gentamicin once daily for 2 days plus oral amoxicillin twice daily for 7 days

- Critical Illness -presence of any of the following signs – convulsions, not able to feed at all and no movement on stimulation.
  - Recommended treatment: Refer URGENTLY to hospital, if referral is still not possible, treatment with daily injectable gentamicin and twice-daily injectable ampicillin until referral is possible or for at least 7 days

**Fig 3. Definitions and treatment for PSBI sub-classifications.** PSBI, possible serious bacterial infection. A syndromic classification for health workers using the IMCI algorithm.

and counseled on the importance of hospital treatment. PHC worker facilitated referral by counseling and providing a referral slip. However, for those who still refused, treatment was offered by the treating PHC worker with the available medicines.

The evaluation for the various outcomes were studied systematically. All infants receiving treatment were assessed for outcomes and documented on the study case record forms. The documentation of challenges was based on team meetings that included field workers, health workers at the facilities, supervisors and health facility managers. It also included field assessments of the health facilities, asking families during follow-up visits to understand the barriers and identify facilitators to implement this intervention

**Commodities.** The required commodities to manage PSBI on outpatient basis were not available through the routine health system at the PHCs. TSU provided the medicines (injection gentamicin, injection ampicillin and oral dispersible amoxicillin tablets) and other commodities such as weighing scales, thermometers and respiratory rate timers.

## Analysis plan

The data collected was analysed using a descriptive analysis using frequencies and proportion of sick young infants identified at different levels (CHW, nurse/doctor), proportion referred to hospital, proportion that did not accept referral, proportion treated at health centre, proportion able to complete treatment, adherence to treatment rates, deaths and other adverse outcomes, proportion that did not improve and needed higher level care, proportion that were cured, proportion that survived, along with information about the barriers and facilitating factors.

## Consent and ethical approval

The implementation research protocol was approved by the ethical review committee of Ahmadu Bello University Teaching Hospital, Zaria and by the WHO Ethical Review

Committee. Written and witnessed informed consent was obtained from parents/caretakers of young infants for agreement to enroll for outpatient management.

## Results

During the study period, a total of 2,154 livebirths were registered by the CHWs at the implementation sites. The CORPs identified and visited 1803(83.7%) newborns on day 1, 2016 (93.6%) were visited on day 3 while 2061(95.7%) were visited on day 7. We believe that the CHWs might have missed up to 25% of all births in the study area, which means that around 2,693 live births could have been identified during the study period at the study site.

### Coverage of PSBI treatment in the study area

During the study period, 347 young infants up to 2 months of age were identified with signs of PSBI, of which 343 received treatment either as an outpatient or in the hospital. Families of four young infants 7–59 days old with fast breathing refused treatment. As the services in the surrounding area PHCs were not optimal, many families started bringing their infants to the study PHCs as staff and medicines were always available (Fig 4). We estimate that around 25% of young infants with PSBI signs came from outside the study area. Thus, we estimate that 257 young infants belonged to the actual study area received PSBI treatment, compared to the expected number of 269 assuming a 10% incidence from the AFRINEST data [9], therefore coverage of PSBI treatment in the study area was 95.5%.

### Identification of sick young infants and acceptance of referral advice

CORPs identified and referred 88.8% (308/347) of the sick young infants who were classified to have PSBI at the PHCs. Nearly all (97.4%, 262/269) families whose young infants were referred to a hospital from a PHC refused to take their infant to the hospital. Therefore, these infants were treated on an outpatient basis. This included 10 families whose young infants had signs of critical illness but they did not want to go to a hospital despite extensive counselling by the PHC staff (Table 2).

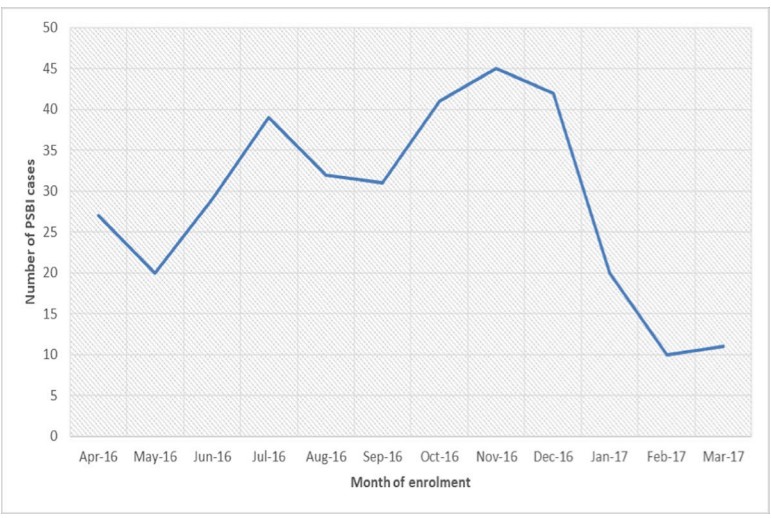

**Fig 4. Identification of PSBI cases at the PHCs by month.**

## Treatment and outcomes of young infants treated with signs of PSBI

All 7–59 days old young infants with fast breathing who received treatment at the outpatient improved with oral amoxicillin (Table 2). Practically all (99%, 340/343) patients were followed up till day 14. Complete adherence to oral amoxicillin treatment was documented in 97.8% (323/327) of those who received it for fast breathing pneumonia or clinical severe infection, and in 99% (147/148) amongst clinical severe infection cases who received all gentamicin doses (Table 2). All 7–59 days old infants with fast breathing pneumonia who were followed up recovered on oral amoxicillin treatment (Table 3). In young infants 0–6 days old with fast breathing 14.4% (15/105) were declared treatment failure, mostly because they had persistence of fast breathing on day 4 of treatment. Only two infants with clinical severe infection failed initial therapy. Ten amongst 327 infants (3.6%) with fast breathing or clinical severe infection died. All of them were treated at the outpatient level (Table 3). Among the 10 infants with critical illness, who refused referral to a hospital were treated with two injections of ampicillin plus one injection of gentamicin daily on an outpatient basis for 7 days, eight recovered and two died (20%, 2/10).

## Challenges and solutions

At the national level dialogue, the paediatricians who were part of the national technical working group were worried about allowing CHEWs to administer oral and injectable antibiotics in the community due to their concern of antimicrobial resistance. Discussions during the

**Table 2. Treatment adherence and follow-up of Young Infants 0–59 days old with signs of PSBI (n = 347).**

| Parameters | 7–59 days FB only n = 78 | 0–6 days FB only n = 105 | 0–59 days CSI n = 154 | 0–59 days CI n = 10 |
|---|---|---|---|---|
| Number identified at the health facility (%) | 78(100) | 105(100) | 154 (100) | 10(100) |
| Number brought directly by families to the health facility (%) | 2(2.6) | 4(3.8) | 33(21.4) | 0(0.0) |
| Number identified by CORPs in the communities and referred to PHC (%) | 76(97.4) | 101(96.2) | 121(78.6) | 10(100) |
| Number refused referral to the hospital (%) | - | 104(99.1) | 148(96.1) | 10(100) |
| Number treated on an outpatient basis at PHCs (%) | 75(96.2) | 104(99.1) | 148(96.1) | 10(100) |
| Number completed treatment (%) (Received 2 injections of gentamicin and all 14 doses of oral DT amoxicillin) | 74(94.9) | 104(99.1) | 145(94.1) | 8(80.0) |
| Adherence to treatment | | | | |
| Number received two injections of gentamicin (%) | - | - | 147(95.5) | 10(100) |
| Number received one injection of gentamicin (%) | - | - | 1(0.7) | - |
| Number received all 14 doses of DT amoxicillin | 74(94.9) | 104(99.1) | 145(94.1) | - |
| Number received 10–13 doses of DT amoxicillin (%) | | | 1(0.7) | - |
| Number received 6–9 doses of DT amoxicillin (%) | | | 2(1.3) | - |
| Number received 5 or less doses of DT amoxicillin (%) | 1(1.3) | | | - |
| Follow up of infants | | | | |
| Number completed all follow-up visits (%) (days 1–7 and 14) | 74(94.9) | 104(99.1) | 154(100) | 8(80) |
| Number partially followed-up (%) (all follow-up visits not completed) | | 1(0.9) | | 2(20) |
| Number lost to follow-up (outcome unknown) (%) | 4(5.1) | | | |

Abbreviations

FB; Fast breathing

CSI; Clinical severe infection

CI; Critical illness

**Table 3. Place of treatment and outcomes for young infants with signs of PSBI (N = 347).**

| PSBI Classification | Hospital treatment | | Outpatient treatment | | |
|---|---|---|---|---|---|
| | Number treated | Deaths with 15 days | Number treated | Clinical treatment failure excluding deaths | Deaths within 15 days of treatment |
| 7–59 days old young infant with fast breathing (n = 78) | 0 (0.0%) | 0 (0.0%) | 74 (94.9%) | 0 (0.0%) | 0 |
| 0–6 days old young infant with fast breathing (n = 105) | 1 (1.0%) | 0 (0.0%) | 104 (99.0%) | 15 (14.4%) | 2 (1.9%) |
| 0–59 days old young infants with signs of Clinical Severe Infection (n = 154) | 6 (3.9%) | 0 (0.0%) | 148 (96.1%) | 2 (1.3%) | 8 (5.4%) |
| 0–59 days old young infants with signs of Critical Illness (n = 10) | 0 (0.0%) | 0 (0.0%) | 10 (100.0%) | 0 | 2(20%) |
| Total (n = 347) | 7 (2.0%) | 0 (0.0%) | 336 (96.8%) | 17 (4.9%) | 12 (3.5%) |

Note: 4 young infants 7–59 days old with fast breathing refused treatment and refused follow up for outcome documentation.

national dialogue and presentations at various professional meetings were held to address their concerns, resulting in their support for the implementation research.

There was a serious challenge of accessibility to a PHC for eight communities in Wucicciri ward as these communities were too far from any functional health centre. There was a general shortage of staff at the PHCs. One of the PHC in Dutsen Abba ward was not functioning, as no staff was posted there. The salaries of the PHC staff had not been paid for some time due to financial difficulties faced by the State government, consequently, the health workers attitude was not patient friendly and the official working hours were not respected and absenteeism was prevalent. There was either a lack of or limited availability of drugs as well as essential equipment such as thermometers, respiratory rate timers and weighing scales at all health facilities. We discussed these challenges with the LGA authorities and the community leaders. The LGA authorities posted one CHEW to the non-operational PHC. To address the challenge of accessibility in Wuciciri ward for eight communities that were too far from the PHC, an adhoc PHC was established in Bogari community with the help of community leaders and LGA health authorities. To help address the poor attitude to work, the TSU provided monthly transport stipends to the CHEWs on government payroll to visit the sick young infants at home when required. The TSU hired six CHEWs and posted one to each PHC in the study area to help fill in the research case report forms and to assist in work due to the shortage of staff in the PHCs. We also undertook a rigorous reorientation of the health workers on general work ethics and responsibilities.

Finding eligible female members of the communities to be engaged as CORPS in view of the practice of women seclusion and restriction of males from entering homes in the absence of the husband was a challenge. However, with series of dialogue and considering the importance of the programme to the communities, the village heads and mai unguwas agreed to use male members of the communities as CORPs allowing them free access to community homes even in their absence.

## TSU support required for scaling up PSBI intervention

As part of scaling up PSBI implementation, Zaria LGA initiated the implementation of management of young infants with PSBI when referral is not feasible in five PHCs of two other political wards Dambo and Kufena, which covered 26 communities. All the nurses/CHEWs in the five PHCs within the two wards were trained and CORPs were identified and trained from within the communities.

The TSU also supported "Hope for the Village Child Foundation" a non-governmental organization to initiate the implementation of management of young infants with PSBI when referral is not feasible in three other LGAs of Kaduna State. Technical support was provided to train all the nurses/CHEWs in these LGAs and to identify the CORPs and train them.

The TSU worked in partnership with national and sub-national policy makers and programme managers to scale up implementation of PSBI within existing structure in other parts of the country. With support from USAID Maternal and Child Survival Programme (MCSP), the TSU supported the Federal Ministry of Health in Nigeria to incorporate the PSBI strategy into the essential newborn care course (ENCC) and the integrated management of childhood illness (IMCI) training materials, the key child survival strategy identified for domestication of the PSBI concept. These two documents have been revised to incorporate the PSBI strategy for country wide scale up (personal communication, Director, Child Health Division, FMoH). Support was also provided to the introduction of PSBI management at PHCs in Kogi and Ebonyi States. A detailed implementation strategy for these states has been developed with drafted indicators for each state.

## Discussion

Our results show that it is feasible to implement the WHO guideline for PSBI treatment when referral is not possible within the existing health structure. We achieved a high treatment coverage of over 95% during the study period. Young infants 7–59 days of age with only fast breathing were safely and effectively treated with oral antibiotic at PHC without being referred to a hospital for the first time. Young infants with clinical severe infection whose parents did not accept referral advice treated at the outpatient level with a combination of oral amoxicillin and intramuscular gentamicin.

The overall case fatality rate (CFR) among young infants with clinical severe infection in this study was 3.6%, which was much lower than the CFR of 9.8% in PSBI cases reported in a systematic review/meta-analysis from LMICs.[25] If these young infants had not been offered this simplified antibiotic therapy when their families refused referral, it is quite likely that many more would have died. Previously published work has shown that prompt and appropriate treatment can reduce CFR in young infants with severe infection by 30–70% [26–27] making it one of the most important strategies for improving newborn survival. Implementation of outpatient treatment when referral is not feasible is critical for low resource settings where majority of families of sick young infants do not accept referral advice and therefore these infants do not get appropriate treatment.[5–9, 28–29]

Experience from Ethiopia also showed that implementing neonatal PSBI management within the existing government health structure through Health Extension Workers at community level was feasible and cost effective.[30]Its inclusion in the intervention areas reduced the post day 1 neonatal deaths by 17% compared to control areas. Also in a similar setting in Nepal, a community-based pilot called- Morang Innovative Neonatal Intervention (MINI) progamme, implemented withing the existing government health infrastructure, showed that Community Health Workers were able to assess and identify possible infections in young infants and delivered appropriate treatment with antibiotics.[31] In the MINI programme, Female Community Health Volunteer (FCHV), who were local women with limited formal education received basic training of 18 days with refresher training worked in their communities to identify sick young infants and refer them to the primary health centre while the Facility Based health workers who were trained and qualified to give intramuscular antibiotics provided treatment with intramuscular gentamicin. However, in the MINI model, the FCHVs initiated treatment in community with oral co-trimoxazole before referral to the primary health

centre. Appropriately trained and supported health workers at the first level health facilities can manage young infants with signs of PSBI with appropriate antibiotics when referral is not feasible as well as for other paediatric infections in resource-limited settings[32] The PHC workers correctly identified and managed sick young infants in line with the guideline on managing infants with PSBI when referral was not feasible. Technical support to them is essential. Since this was the first time a guideline recommending treatment of sick young infants with injectable therapy on an outpatient basis was implemented, apprehension on the part of even experienced health workers was natural. The TSU provided the technical support, did hand-holding and built their confidence in managing the sick young infants. We acknowledge that it may not be feasible to establish TSUs throughout the country when this intervention is scaled up. But it is imperative to have some kind of technical support available at district level through a linkage between the PHCs and their feeder hospital paediatricians and/or neonatologists to provide mentorship and solve problems. There is concern about quality assurance for safe injectable antibiotics by health workers.[8]No adverse event such as an injection abscess was reported during the study period attesting to the fact that appropriately trained and supervised health workers can safely deliver such treatment to young infants with PSBI at the first level health facilities. The is similar to experience from Nepal who reported no adverse event throughout the course of the MINI programme.[30]

Scale-up and implementation of PSBI treatment guideline requires government commitment and motivation of the health facility staff. If that commitment is not there in the shape of human and financial resources, this will not happen. The input of TSU to address health system challenges identified above was a critical element in achieving high coverage of treatment with quality. We believe it is possible for the authorities to address the challenges of accessibility to PHCs and the inadequate number of health staff in the near future by engaging additional CHEWs from the pool of unemployed CHEWs within Zaria. The health facility staff need to be paid regularly on time to keep their motivation, so that they carry out their work diligently. In line with the State government policy the PHCs should also have stock of essential medicines and supplies needed to provide maternal and child health services.

Our implementation research had several strengths. First, we tried to implement the PSBI strategy within the existing structure; the primary health facilities and staff dispensation. Secondly, the capacities of the CHEWs employed as facility health workers was improved upon through appropriate training and supervision to position them for effective health care delivery. Finally, we used members of the communities as CORPs and through this process institutionalized the CORPs system and stimulated health programme ownership in these communities.

One potential limitation in our study was the close supervision by the TSU in follow-up of patients resulting in high coverage and treatment success, which may not be possible to replicate in real life program setting. Second, the TSU filled in the gaps that were the responsibility of the LGA such as provision of medicines and commodities which if not done, would have resulted in no implementation of this intervention. Third, TSU hired personnel like CORPs to identify births and sick young infants in the community and the CHEWs hired to collect research related data also assisted in the routine PHC work in the absence of unpaid staff who were demotivated to work regularly and diligently.

In conclusion, WHO PSBI guideline when referral is not feasible is implementable within the programmatic setting in Nigeria and can help achieve high treatment coverage. Delivery of treatment for young infants with PSBI has great potential to address the high neonatal mortality especially in the LMICs. However, the implementation will require commitment of government to put in place a functional health system with adequate medicines and supplies in health facilities, adequate number of trained, motivated and paid health workers.

## Supporting information

**S1 Appendix. Dataset.**
(ZIP)

## Author Contributions

**Conceptualization:** Shamim Ahmad Qazi, Samira Aboubaker, Rajiv Bahl.

**Formal analysis:** Robinson Daniel Wammanda, Yasir Bin Nisar, Shamim Ahmad Qazi, Samira Aboubaker, Rajiv Bahl.

**Investigation:** Shamim Ahmad Qazi.

**Methodology:** Robinson Daniel Wammanda, Shamim Ahmad Qazi, Samira Aboubaker, Rajiv Bahl.

**Project administration:** Robinson Daniel Wammanda, Shadrach Aminu Adamu, Hyellashelni Daba Joshua.

**Supervision:** Shadrach Aminu Adamu, Hyellashelni Daba Joshua, Shamim Ahmad Qazi, Samira Aboubaker, Rajiv Bahl.

**Writing – original draft:** Robinson Daniel Wammanda, Shadrach Aminu Adamu, Hyellashelni Daba Joshua, Yasir Bin Nisar, Shamim Ahmad Qazi, Samira Aboubaker, Rajiv Bahl.

**Writing – review & editing:** Robinson Daniel Wammanda, Yasir Bin Nisar, Shamim Ahmad Qazi, Samira Aboubaker, Rajiv Bahl.

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
