## [Decision Letter · Decision Letter 0]

25 Nov 2019

PONE-D-19-27113

Implementation of the WHO guideline on treatment of possible serious bacterial infection when hospital referral is not feasible in Zaria, Nigeria: Challenges and solutions

PLOS ONE

Dear Dr Wammanda,

Thank you for submitting your manuscript to PLOS ONE. After careful consideration, we feel that it has merit but does not fully meet PLOS ONE’s publication criteria as it currently stands. Therefore, we invite you to submit a revised version of the manuscript that addresses the points raised during the review process.

Specific comments are provided below, but we are requesting a more detailed description of methods particularly with regards to collection of data. We also recommend adding graphics (tables or figures), where possible, to better communicate some of the methods and results. 

We would appreciate receiving your revised manuscript by Jan 02 2020 11:59PM. To enhance the reproducibility of your results, we recommend that if applicable you deposit your laboratory protocols in protocols.io, where a protocol can be assigned its own identifier (DOI) such that it can be cited independently in the future. For instructions see: http://journals.plos.org/plosone/s/submission-guidelines#loc-laboratory-protocols

We look forward to receiving your revised manuscript.

Kind regards,

Surbhi Leekha

Academic Editor

PLOS ONE

Journal Requirements:

Additional Editor Comments (if provided):

1. Given that the study was designed and presented as "implementation research, please specify if a formal implementation model or framework was used, and cite appropriate implementation science literature if relevant.

2. Was there a specific reason to select Zaria LGA as the study site?

3. The study describes both development of the implementation, evaluation of outcomes, as well as some description of challenges and solutions within the implementation process – please state explicitly whether these were studied systematically or based on anecdotes, team meetings etc. For example, were specific questions asked of field workers periodically to understand barriers? This should be included in the methods.

4. The Methods section "Evaluation of existing PHCs, health workers and commodities" includes several results of those evaluations: those results are better suited for the results, particularly as additional results of that evaluation are described in the Results section under Challenges and Solutions e.g., general shortage of staff at PHCs. Alternatively, if you prefer to keep some of this is the methods section, state explicitly that this was pre-implementation evaluation to inform the intervention implementation, and tie it into the paragraph that immediately follows (establishment of TSU, and how this evaluation informed what the TSU needed etc).

5. Please limit the use of abbreviations and acronyms where possible e.g., TBA acronym is not necessary. Provide reference for Integrated Community Case Management (iCCM) (acronym not necessary as it is not used again)

6. In the discussion, it is stated that “No adverse event such as an injection abscess was reported..” Were these data collected systematically or is this just anecdotal?

7. Consider adding some discussion on how the feasibility and implementation experience compares with similar interventions in other LMIC.

Reviewers' comments:

Reviewer's Responses to Questions

**Comments to the Author**

1. Is the manuscript technically sound, and do the data support the conclusions?

Reviewer #1: Partly

Reviewer #2: Yes

2. Has the statistical analysis been performed appropriately and rigorously? 

Reviewer #1: N/A

Reviewer #2: Yes

3. Have the authors made all data underlying the findings in their manuscript fully available?

Reviewer #1: No

Reviewer #2: Yes

4. Is the manuscript presented in an intelligible fashion and written in standard English?

Reviewer #1: No

Reviewer #2: Yes

5. Review Comments to the Author

Reviewer #1: This manuscript by Wammanda and colleagues describes the adaptation and multisite implementation of the World Health Organization’s (WHO) guidelines for the treatment of possible serious bacterial infection (PSBI) in rural northern Nigeria. This report in important because it describes the context-specific adaptation of a guideline that has broad application and the potential to impact that outcomes of a significant number of neonates; describes a novel approach to guideline dissemination and implementation in a rural setting where organized healthcare settings are sparse, and in a Muslim population in which female seclusion is the norm.

While the topic and work is very important, the manuscript could be substantially improved to ensure it communicates effectively the work that was done and shares with others the implementation plan that was used.

MAJOR COMMENTS:

1) Some of the work described to implement this guideline in rural northern Nigeria included working with the national MOH and NGOs. I did not find a description of how more local public health and child health experts were engaged.

2) Additional information would be valuable about how the specific communities within Zaria were chosen for this implementation project.

3) More definitions are needed for the data elements that were captured and the way in which data were collected. This is particularly important for adherence to treatment, other adverse outcomes, etc.

4) The description of the study population is vague. To describe the impact of this intervention, it would be very valuable to have a more precise way to define and identify both patients that were in the population cohort as well as children identified as those members of the cohort who were identified as having PSBI. Was this a birth cohort in which babies were followed to 2 months of age or event (which ever came first)? How were members of the cohort identified (or numbers of members estimated) during the study period? Did all of this occur via registration of pregnant women by CORPs staff? Can you estimate the percentage of live births in the region that were captured by this strategy?

5) Ascertainment of the babies that experienced PSBI is subject to bias, particularly for the families in the most remote areas understudy. Can the authors provide us some data on the proportion of live births who had the anticipated home visits (days 1, 3,7) by CORP workers?

6) The authors seem to have retained infants identified with PSBI from regions outside of the study area in the group on which they report outcomes. These babies should be removed.

7) The authors do an excellent job describing the social context in which they were working. However, it would be valuable to provide a bit more information about the economic context and penetration of other health initiatives. What were the professions of most people in these commerunities (e.g., traders, subsistence farming)? What was the coverage of health interventions such as basic vaccination?

8) The authors provide a detailed description of the micro-economics of purchase and replenishment of antibiotics in the primary health centers as well as the fact that care provided in those sites was free. I am confused about what costs a family would have to bear to bring their child to a PHC, get antibiotics if prescribed, fund transport to a hospital if referred, and pay for hospital-based care if it is recommended. These economic barriers to guideline adherence should be described succinctly since this novel program that provides an alternate to hospital referral may be more attractive to families simply based on cost.

9) Some of the information provided in both the methods and results might be more effectively and efficiently communicated with use of figures and/or tables. Examples include 1) phases of implementation (figure),

10) It seems as though there were multiple types of trainings developed and delivered. However, this is not clearly described. Please clarify and include who did the trainings, how the adequacy of training was assessed, and provide materials in supplement if possible.

OTHER COMMENTS:

1) The title does not contain several important pieces of information – that this is a pediatric study population and that a major part of the work described is about the development of a context-specific implementation plan. It may also be good to indicate in the title that this is about the implementation in a rural setting, since others may search to find reports of how to overcome the unique challenges associated implementation of health interventions where formal healthcare facilities are sparse.

2) The authors could use some assistance with correcting basic errors of syntax, which would improve the capacity of readers to comprehend some aspects of their work. Also need to standardize their approach to introduction and use of abbreviations.

3) Several phrases are unclear to me: “patent medicine vendor”, “seed drugs”

4) I had a hard time keeping straight the various types of healthcare workers (CHEWs, CORPs, CHO, etc). It might be good to provide a glossary or table that pulls together all of these abbreviations, definitions, setting in which they practice, and the scope of their practice.

5) What tool was used to “assess the status of health facilities” to determine capacity of existing resources to implement this guideline? Is this something that could be provided as part of supplementary materials?

Reviewer #2: 1. The authors should include a detailed description of how training to lay people for them to administer IM injections to infants at appropriate dosages was done. This should include needle stick safety and disposal.

2. How rigorous was the adverse event recording documentation after IM injections?

3. The authors should include under "limitations" why they did not take blood cultures to validate the claim for bacterial infections in this population. Antimicrobial sensitivity would have guided in future antibiotic use and this could have provided baseline data

4. Figure 1 seems to suggest seasonal variation of infection. What are the RSV rates in this population? Viral infection usually precedes bacterial infections.

6. PLOS authors have the option to publish the peer review history of their article (what does this mean?). If published, this will include your full peer review and any attached files.

Reviewer #1: No

Reviewer #2: No

---

## [Author Response · Author response to Decision Letter 0]

16 Jan 2020

Response to reviewers and editor provided

---

## [Editor Report · Decision Letter 1]

23 Jan 2020

Implementation of the WHO guideline on treatment of young infants with signs of possible serious bacterial infection when hospital referral is not feasible in rural Zaria, Nigeria: Challenges and solutions

PONE-D-19-27113R1

Dear Dr. Wammanda,

We are pleased to inform you that your manuscript has been judged scientifically suitable for publication and will be formally accepted for publication once it complies with all outstanding technical requirements.

With kind regards,

Surbhi Leekha

Academic Editor

PLOS ONE
---

## [Editor Report · Acceptance letter]

2 Mar 2020

PONE-D-19-27113R1 

Implementation of the WHO guideline on treatment of young infants with signs of possible serious bacterial infection when hospital referral is not feasible in rural Zaria, Nigeria: Challenges and solutions 

Dear Dr. Wammanda:

I am pleased to inform you that your manuscript has been deemed suitable for publication in PLOS ONE. Congratulations! Your manuscript is now with our production department. 

With kind regards,

on behalf of

Dr. Surbhi Leekha 

Academic Editor

PLOS ONE